# Effects of Rich-Polyphenols Extract of Dendrobium loddigesii on Anti-Diabetic, Anti-Inflammatory, Anti-Oxidant, and Gut Microbiota Modulation in db/db Mice

**DOI:** 10.3390/molecules23123245

**Published:** 2018-12-07

**Authors:** Xue-Wen Li, Hui-Ping Chen, Ying-Yan He, Wei-Li Chen, Jian-Wen Chen, Lu Gao, Hai-Yan Hu, Jun Wang

**Affiliations:** 1School of Pharmaceutical Sciences, Sun Yat-sen University, Guangzhou 510006, China; lixw28@mail2.sysu.edu.cn (X.-W.L.); chenhp25@mail2.sysu.edu.cn (H.-P.C.); heyingy2@mail2.sysu.edu.cn (Y.-Y.H.); wailap_1993@hotmail.com (W.-L.C.); cjw1975@263.net (J.-W.C.); 2BGI-Shenzhen, Shenzhen 518083, China; gaolu@bgi.com

**Keywords:** *Dendrobium loddigesii*, rich-polyphenols extract, db/db mice, type 2 diabetes mellitus, diabetes complication, inflammation, oxidative stress, gut microbiome

## Abstract

*Dendrobium* is a traditional Chinese herb with anti-diabetic effects and has diverse bibenzyls as well as phenanthrenes. Little is known about *Dendrobium* polyphenols anti-diabetic activities, so, a rich-polyphenols extract of *D. loddigesii* (DJP) was used for treatment of diabetic db/db mice; the serum biochemical index and tissue appearance were evaluated. In order to gain an insight into the anti-diabetic mechanism, the oxidative stress index, tumor necrosis factor-α (TNF-α), interleukin-6 (IL-6) and gut microbiota modulation were determined by ELISA, immunohistochemistry or high throughput sequencing 16S rRNA gene. The results revealed that DJP had the effects to decrease the blood glucose, body weight, low density lipoprotein cholesterol (LDL-C) levels and increase insulin (INS) level in the mice. DJP improved the mice fatty liver and diabetic nephropathy. DJP showed the anti-oxidative abilities to reduce the malondialdehyde (MDA) level and increase the contents of superoxide dismutase (SOD), catalase (CAT) as well as glutathione (GSH). DJP exerted the anti-inflammatory effects of decreasing expression of IL-6 and TNF-α. After treatment of DJP, the intestinal flora balance of the mice was ameliorated, increasing *Bacteroidetes* to *Firmicutes* ratios as well as the relative abundance of *Prevotella*/*Akkermansia* and reducing the relative abundance of *S24-7*/*Rikenella*/*Escherichia coli.* The function’s prediction of gut microbiota indicated that the microbial compositions involved carbohydrate metabolism or lipid metabolism were changed. This study revealed for the first time that DJP improves the mice symptoms of diabetes and complications, which might be due to the effects that DJP induced the decrease of inflammation as well as oxidative stress and improvement of intestinal flora balance.

## 1. Introduction

Type 2 diabetes mellitus (T2DM) is a growing threat to public health. International Diabetes Federation estimated that in 2017 there were 451 million (age 18–99 years) people with diabetes worldwide, and the people with diabetes were expected to increase to 693 million by 2045 [1]. T2DM is a metabolism correlation disease; its main symptoms are hyperglycemia, insulin resistance, and decrease of insulin secretion. T2DM can cause death and disability because of various severe, whole-body complications in organs, such as, heart, kidneys, liver, brain, and others. 

The T2DM and complications etiology is complexity, many factors influencing their onset and development, such as, environmental and genetic factors. In recent years, oxidative stress and inflammation were recognized as one of the important causes of T2DM and complications [2]. Oxidative stress was considered as a unifying mechanism of diabetic and its complications [3]. Diabetes can lead to abnormal metabolism of energy, carbohydrates, fat and protein [4,5,6]; high glucose and free fatty acids levels can stimulate oxidative stress, increase the expression of inflammatory factors and lead to insulin resistance [7]. IL-6 and TNF-α, as major inflammatory factors, participate in the development of T2DM; the increasing expression of TNF-α and IL-6 can cause pathological injury and diabetes complications [8,9]. 

In recent years, gut microbiota was regarded as an important contributing factor in the occurrence and developing of T2DM [10,11]. The symbiosis of intestinal flora with humans is thought of as a “vital organ” of the human body, which can affect the host’s energy metabolism; the studies of germ-free mice provided the evidence that the diversity or relative proportion of gut microbes played active roles in energy homeostasis [12]. The altered gut microbiota could affect inflammatory response as well as insulin resistance and play an important role in the development of T2DM [13]. Modulation of gut microbiota may become a novel therapeutic approach for T2DM [14].

“Shi-Hu” is a traditional Chinese medicine herb and from different species of *Orchidaceae Dendrobium* [15], which contains many compounds, such as polyphenols, polysaccharides, and alkaloids [16]. *Dendrobium* polyphenols are mainly bibenzyls, phenanthrenes, or their polymers [16]. *Dendrobium* polyphenols have diverse bioactivities [17,18], such as, moscatilin, a dominant component of *Dendrobium* polyphenols, was reported to have antiplatelet aggregation, antimutagenicity, inducing apoptosis of cancer cells, and anti-inflammatory activity [19,20,21,22]; some polymers of bibenzyls or phenanthrenes showed inhibiting α-glucosidase activity [23]. *Dendrobium* plants have been used to treat T2DM and complications in traditional Chinese medicine clinic [24,25], such as, ethanol extract of *D. chrysotoxum* showed the abilities to ameliorates diabetic retinopathy [26]. *D. loddigesii* is one of the most important “Shi Hu” crude drugs in traditional Chinese medicine clinic, which contains abundant *Dendrobium* polyphenols besides polysaccharides and alkaloids [27,28]. In a previous study we found that the rich-polyphenols extract of *D. loddigesii* (Appendix A) and its phenols components showed strong anti-inflammation and antioxidant activities in vitro [29]. To study anti-diabetic effects of *Dendrobium* polyphenols, a rich-polyphenols extract of *D. loddigesii* was prepared and its anti-diabetic effects, anti-oxidant, anti-inflammatory activities and modulation of gut microbiota were studied using db/db mice. 

## 2. Results

In Chinese medicine, prescript uses *Dendrobium* plants to treat T2DM and diabetes complications [24,30]. In order to explore the anti-diabetic effects of *Dendrobium* polyphenols, a rich-polyphenols extract (DJP) was prepared from *D. loddigesii* and used for treatment of diabetic db/db mice (BKS.Cg-Dock7m +/+Leprdb/Nju) in 3 dosage groups, DMDJP25 group (in a dose of 25 mg/kg), DMDJP50 group (in a dose of 50 mg/kg), and DMDJP100 group (in a dose of 100 mg/kg). BKS-Leprdb/Leprdb (BKS-db) mice are a widely used for T2DM animal models. Mice homozygous for the diabetes spontaneous mutation (Leprdb) manifest morbid obesity, hyperglycemia, diabetic nephropathy, steatohepatitis, and heart disease [31]. After treating 8 weeks, the effects of DJP on blood picture and organ histomorphology in the treated mice were observed and compared with no-treatment db/db mice group (DM group), control group (C57 group), and metformin (MET) treatment group (DMMET130 group, in a dose of 130 mg/kg) was used as a positive control group. To understand the anti-diabetic effect of DJP, the oxidative stress parameters, inflammatory factors, and modulation of gut microbiota were studied in the mice.

### 2.1. Main Compounds in DJP

DJP is an enriched polyphenols fraction, from which about 21 phenols were identified using chromatographic techniques [27,28]. Under our analytical conditions only four components, moscatilin (**1**), gigantol (**2**), 2,4,7-trihydroxyl-9,10-dihydro-phenanthrene (**3**), and tristin (**4**) (Appendix A), were evaluated, which concentrations were 7.2%, 1.9%, 0.62%, and 0.23%, respectively.

### 2.2. Anti-Diabetic Effects of DJP

#### 2.2.1. Effects of DJP on the Body Weight, Blood Glucose Level, and Oral Glucose Tolerance (OGTT)

The body weight and blood glucose level of the mice were tested once a week; the results were showed in Figure 1. OGTT was tested at the 7th week; the result was showed in Figure 2. 

Figure 1A showed that the body weight of db/db mice was heavier than that of C57 control mice by 128–98.7%. There was no obvious difference of the body weight among DM, DMDJP25, and DMMET130 groups. After 7 weeks of treatment, the body weight of DMDJP50 and DMDJP100 were significantly decreased by 15% and 13%, respectively. (*p* < 0.05 vs. DM group). 

Figure 1B showed that as DJP concentrations increases up to DMDJP100 the blood glucose level enhanced to 21.0 mmol/L compared to C57 control mice (8.3 mmol/L). Figure 2 displayed that the OGTT test in the db/db mice indicated that all DJP concentrations mice showed higher OGTT areas than that of C57 control mice by enhancing blood glucose level by 120%. However, the level of blood glucose and the OGTT test in db/db mice were not affected with the concentrations of DMDJP25 and DMDJP50 tested. DMMET130 group enhanced the glucose tolerance level by 19.3% in OGTT test after 6 weeks of treatment (*p* < 0.05) compared to DM group. DMDJP100 group enhanced the glucose tolerance level in OGTT test (*p* < 0.05 vs. DM group) after 6 weeks of treatment, which was similarly to the effect induced by DMMET130 group.

#### 2.2.2. Effect of DJP on Serum Insulin and Lipid Levels 

The concentrations of triglyceride (TG), total cholesterol (TC), high density lipoprotein cholesterol (HDL-C), LDL-C, and INS in the mice serum were determined after the animal experiment; the results are shown in Table 1.

Table 1 indicates that DMDJP25, DMDJP50, and DMDJP100 treatments of db/db mice enhanced the concentration of TG in serum by approximately 1.50 mmol/L compared to control C57 mice (1.01 mmol/L). In addition, in the same experiments TG, TC, and HDL-C levels were not affected with different doses of DJP or MET (in 130 mg/kg dose). Further experiments in the same mice with DMDJP50 and DMDJP100, the level of LDL-C decreased than DM group (*p* < 0.05) by 0.69 and 0.65 mmol/L, which were not decreased with DMDJP25 and DMMET130 treatment. Furthermore, the concentration of INS in DM group was lower by 7.06 mIU/L than control group or DJP treatment. Finally, the serum INS level in the db/db mice treated with DMDJP50, DMDJP100, and DMMET130 increased by 10.93, 11.47, and 11.18 mIU/L than DM group 7.06 mIU/L (*p* < 0.05). 

#### 2.2.3. Effect of DJP on Tissue Forms of Liver/Kidney 

The changes of tissue forms of liver/kidney in the mice were observed by HE staining, and the results were showed in Figure 3 and Figure 4, respectively; the result of pathological lesion analysis were displayed in Table 2. 

The liver tissues in C57 control group showed normal morphology and a few lymph cell infiltrations were found occasionally. The liver tissues in DM group displayed severe hepatic steatosis accompanied by lymph cell infiltration. Compared with DM group, the hepatic steatosis in DMMET130 group was relieved (Figure 3B and Table 2). After 8 weeks of DJP treatment, the hepatic steatosis in DMDJP25 group was not improved, but the hepatic steatosis were significantly decreased in DMDJP50 group (*p* < 0.05 vs. DM group) and DMDJP100 group (*p* < 0.01 vs. DM group), which displayed certain dosage-effect relationship (Table 2). The lymph cell infiltration of liver disappeared in DMDJP100 group.

The kidney tissues in C57 control group was normal morphology. The renal tubule epithelium cells in DM group showed varying degrees of vacuolar degeneration; the renal lesion in DMDJP25 group and DMDJP50 group were not improved. The vacuolar degeneration of renal tubular epithelial cells was not observed in DMMET130 and DMDJP100 groups (Figure 4B,F and Table 2; *p* < 0.01 vs. DM group).

### 2.3. Effect of DJP on Inflammatory Factors 

To understand the anti-diabetic effect of DJP, the level of serum IL-6 was detected by ELISA and the level of TNF-α in the liver/kidney tissue were detected by immunohistochemistry. The results were showed in Figure 5, Figure 6, Figure 7 and Table 3, respectively. 

Figure 5 revealed that the level of serum IL-6 in the treatment groups was lower than that of DM group; the three groups of DJP treatment displayed certain dosage-effect relationship. Compared with DM group, the level of serum IL-6 was significantly reduced in DMDJP100 group (*p* < 0.05).

Figure 6, Figure 7, and the gray-value in Table 3 showed that the relative expression amount of TNF-α in control group was the lowest level, while that in DM group had the highest level. The relative expression amount of TNF-α in DMDJP100 group was significantly reduced (*p* < 0.05 in kidney tissue; *p* < 0.01 in liver tissue, vs. DM group). 

### 2.4. Effect of DJP on Oxidative Stress Index 

To understand the anti-diabetic effect of DJP, SOD, MDA, CAT, and GSH levels of kidney/liver tissue were detected by ELISA. The results were showed in Table 4. 

Table 4 revealed that the MDA of liver/kidney tissue were at high level in DM group, but GSH, CAT, and SOD were at low levels. After administration of DJP or MET, the MDA level were significantly decreased in DMDJP100 and DMMET130 groups; the GSH, CAT, and SOD levels of liver/kidney tissue were significantly increased.

### 2.5. DJP on Modulation of Intestinal Microbiome

#### 2.5.1. Taxonomic Characterization of Intestinal Microbiome

A total of 1,383,220 high-quality sequences from 46 intestinal microbiome samples were produced and with an average of 30,070 sequences per sample. The high-quality sequences were delineated into 607 operational taxonomic units (OTUs) at 97% similarity. The detailed characteristics of each sample were listed in Appendix A. The sequencing quality was evaluated by methods of rarefaction analysis based alpha diversity indexes (Figure 8). From Figure 8, we can see three rarefaction curves (Figure 8A chao1 index curve, 8B Shannon index curve and 8C Simpson index curve) tend to be flat or reach the plateau stage, which suggest the sequencing data were enough to cover all species in the microbiome community, and the most of the gut microbial diversity in each sample was captured with the current sequencing depth. 

Ten different bacterial phyla were identified in the mice intestinal microbiome. The majority of sequences obtained belonged to *Bacteroidetes* (46–61%), *Firmicutes* (26–31%), and *Proteobacteria* (5–16%), followed by *Verrucomicrobia*, *Deferribacteres*, *Cyanobacteria*, *Actinobacteria*, *Tenericutes*, and *TM7*; unclassified bacteria were less 0.03% (Figure 9A). This composition was consistent with previous observations [32,33]. 

Figure 9B showed that the class *Bacteroidia* (61–46%) and *Clostridia* (24–33%) were the most abundant. The genera *Oscillospira* (5%) and *Prevotella* (4%) were the dominated microbiome in control group, and the genera *Prevotella* (11–26%) and *Bacteroides* (8–18%) were the predominance microbiome in the five db/db mice groups. The genera *Oscillospira*, *Prevotella*, *Akkermansia,* and *Mucispirillum* were enriched in control group, whereas *Prevotella* and *Bacteroides* were enriched in the five groups of db/db mice (Figure 9B,C). The genera *Escherichia* was enriched in DM group. A total of 11 species were detected in all 46 samples, which existed large differences of dominant species among the six mice groups (Figure 9C). 

#### 2.5.2. Distinct Microbial Composition

The box-plot based on alpha diversity (Figure 10A) and the principal component analysis (PCA) based on the OTUs abundance (Figure 10B) displayed the differences of microbial composition among the six groups. The distribution of OTUs abundance was similarity between DMMET130 and DMDJP100 groups; the distribution of abundance in DM group was the same as that of DMDJP25 group; the abundance of microbial composition in control group was separated from the five db/db mice groups for mutated gene.

#### 2.5.3. Functional Predictions of Gut Microbiome

The taxa predicted by 16S RNA marker gene sequencing in the six groups were functionally assigned to KEGG (Kyoto Encyclopedia of Genes and Genomes) pathways, and the proportion of sequences (%) were assessed. A total of 328 pathways were detected in the six groups making up the core metabolic functions in the cohort. Overall, the six groups tended to share more similar abundances of sequences families. About 50% of the gut microbiome in almost every group were enriched on KEGG metabolism pathways, such as, carbohydrate metabolism (about 10%), amino acid metabolism (about 9%), energy metabolism (about 6%), metabolism of cofactors and vitamins (about 4%), nucleotide metabolism (about 4%), glycan biosynthesis and metabolism (about 3%), lipid metabolism (about 3%), and enzyme families (about 2%). Using a *p* < 0.05 for ANOVA tests in STAMP, several statistical differences were found between DMDJP100 group and DM group (Appendix A). 

## 3. Discussion

T2DM was called “Wasting and Thirsting Disorder” in ancient Chinese medicine. Shi-Hu, as a “nourishing Yin and invigorating Qi” herb, is the main component of some prescriptions for diabetes [34]. In recent years, the pharmacology and mechanism of *Dendrobium* for diabetes attracted many researchers; however, much attention of researchers were paid on the ethanolic/water extracts, polysaccharide or alkaloid of *Dendrobium* [26,34]. The present study was the first work on the effects of anti-diabetic and improving complications of *Dendrobium* rich-polyphenols extract. 

T2DM complications, such as nephropathy, adiposis hepatica, hypertension, stroke, or ophthalmopathy are some of the causes of high mortality. Some antioxidants and anti-inflammatory agents were used for treatment of diabetes and its complications [35]. Brasnyó et al. reported that resveratrol improved insulin sensitivity in T2DM due to decrease in oxidative stress, expression of NF-ĸB, JNK, TNF-α, and IL-6 [36]. It was reported that curcusome had renoprotective effects by inhibiting renal lipid accumulation and oxidative stress through AMPK and Nrf2 signaling pathway [37]. DJP (Appendix A) and its phenols components showed anti-inflammation and antioxidant activities in vitro [29]; the present study demonstrated that DJP (in a dose of 100 mg/kg) can significantly decrease the blood glucose, LDL-C, insulin resistance and increase serum INS level in db/db mice. In addition, DJP obviously alleviated the fatty degeneration of liver cells and vacuolar degeneration of renal tubule epithelium cells in the mice. ELISA and immunohistochemical analysis revealed that DJP induced the decreasing oxidative stress, IL-6 and TNF-α. It is supposed that the anti-inflammatory and antioxidant activities might be a mechanism for DJP anti-diabetes action.

The intestinal microbiome consists of a complex microbes community that impact normal physiology and susceptibility to disease mediated by inflammatory molecules such as lipopolysaccharides and peptidoglycans [38,39]. It was reported that the intestinal flora of diabetic patients are significant difference from non-diabetic adults [40]; it has been verified that the *Bacteroidetes* to *Firmicutes* ratio was decreased in T2DM individuals [41]. The decrease in *Bacteroidetes* and the increase in *Firmicutes* could be related with the presence of genes encoding enzymes that break down polysaccharides, which was associated with the increased capacity to harvest energy from food, and caused low-grade systemic inflammation [42,43]. In db/db animal model, the *Bacteroidetes* and *Firmicutes* ratio was significant reduction [33]. In this study, we observed that *Bacteroidetes* to *Firmicutes* ratio was increased in DMDJP100 group compared with DM group (Figure 9A), which revealed that the intestinal flora imbalance was improved after 8 weeks of DJP treatment. 

It was reported that the abundance of *Akkermansia*. *muciniphila* was decreased in obese and T2DM mice [44]. Several different research groups confirmed that the increased relative abundance of *Akkermansia* sp. may be a new anti-hyperglycemic mechanism of metformin [45,46]. In the present work, we observed that the relative abundance of *Akkermansia muciniphila* was increased in DMDJP100 group as that of DMMET130 group, which was almost 24 times more than the level of DM group (Figure 11). It is suggested that the increase of relative abundance of *Akkermansia* sp may contribute to DJP anti-diabetic effect.

It was reported that *Escherichia coli* was increased in both obese and T2DM subjects [47]. *Escherichia coli* can produce a material very similar to insulin, which was found to block insulin by binding to the target cell of insulin and leads to diabetes [48]. *Escherichia coli* are intestinal mucosal adherence, and associated with a low-grade inflammation [49]. In present work, the relative abundance of *Escherichia* was significantly decreased in DMDJP100 group (*p* < 0.05 vs. DM group; Figure 11, Figure 12 and Appendix A). It was suggested that DJP improved the mice gut health by gut microbiota modulation, which might be a mechanism for DJP anti-diabetes action.

In this study, we identified several compositional variations of intestinal microbiome (Figure 11, Figure 12 and Appendix A), such as, the relative abundance of *Prevotellaceae* (*Bacteroidetes*) was significantly increased in DM DJP100 group; the relative abundance of *S24-7* (*Bacteroidetes*) and *Rikenella* (*Bacteroidetes*) were significantly decreased (*p* < 0.05 vs. DM group). Monk et al. reported that *Prevotella* and *S24–7* are carbohydrate fermenting and short chain fatty acid (SCFA) production bacteria [50]; SCFA can enhance the barrier function of colonic epithelium [51]. *Rikenella* was sulfatase-secreting bacteria [52], which induced the increase levels of bacterial endotoxinemia and chronic low-grade inflammation [53]. It was suggested that the changes of intestinal flora decreased inflammation and enhanced the mucus/epithelial barrier integrity in DMDJP100 group.

Functional annotation analyses showed that some proportion of sequences involved in individual metabolic pathways in DMDJP100 group are significantly different compared to DM group (*p* < 0.05, Appendix A), such as, the significant decline of bacteria responsible for butanoate metabolism (carbohydrate metabolism)/fatty acid metabolism (lipid metabolism), and the significant increase of bacteria responsible for fatty acid biosynthesis (lipid metabolism)/arachidonic acid metabolism (lipid metabolism) in DM DJP100 group (*p* < 0.05 vs. DM group), which indicated that DJP induced an important change of carbohydrate metabolism and lipid metabolism by the modulation of the gut microbiota.

The functional annotation analyses allowed to identify associated pathways and therefore can predict functional capabilities, but its limitations are the small sample size. To further confirm functional differences before and after treatment, quantification of messenger RNA and metabolic profiling are needed. In spite of this, our study suggested that the changes of intestinal microbiome may markedly contribute to DJP anti-hyperglycemic effects.

## 4. Materials and Methods 

### 4.1. Materials and Reagents

The dried stems of *D. loddigesii* (from Yunnan Province, China) were purchased in September 2015 from Caizilin Pharmacy in Guangzhou, China and identified with the classical method by pharmaceutical botanist Prof. Lin Jiang, School of Pharmaceutical Sciences, Sun Yat-Sen University. The voucher sample (No. 20150913l) was deposited in the School of Pharmaceutical Sciences, Sun Yat-Sen University (Guangzhou, China). The reference compounds including moscatilin, gigantol, 2,4,7-trihydroxyl-9,10-dihydrophenanthrene, and tristin were isolated previously [27]. 

TNF-α antibody and horseradish peroxidase (HRP) conjugated anti-mouse IgG were purchased from Wuhan Servicebio Technology Co., Ltd (Wuhan, China). E.Z.N.A.^®^ Bacterial DNA Kit was purchased from Omega Bio-tek (Norcross, GA, US); ELISA kits were purchased from Nanjing Jiancheng Institute of Bioengineering (Jiangsu, China). Metformin hydrochloride and sodium carboxymethylcellulose (NaCMC) were purchased from Sigma Chemical Co. (St. Louis, MO, USA); Chemical reagents were purchased from Guangzhou Chemical Reagent Factory (Guangzhou, China).

### 4.2. Preparation Rich-Polyphenols Extract of D. Loddigesii 

The dry stems of *D. loddigesii* (1 kg) were extracted with acetone-water solution (80:20, *v/v*; 3 × 5L) at room temperature to generate 70 g of crude extract. The crude extract was dissolved in methanol, deposited on a column of RP-18 gel, and eluted with water, then, water-methanol solution (15:85, *v/v*). The water-methanol eluate was evaporated to dryness to give the rich-polyphenols extract of *D. loddigesii* (28 g), 2.8% of the weight of raw materials. 

The rich-polyphenols extract of *D. loddigesii* (DJP) was subjected to liquid chromatography fingerprint analysis in which major peaks were identified as the marker compounds of their originating herbs [27]; the rich-polyphenols extract of *D. loddigesii* was used in the current pharmacologic study.

### 4.3. Animal Experiments

Eight mice of C57BL/6 (male, 6–8 weeks old) and forty mice of BKS.Cg-Dock7m +/+Leprdb/Nju (db/db mice, male, 6–8 weeks old) were purchased from Nanjing Biomedical Research Institute of Nanjing University (Approval No. SCXK (SU) 2015-0001). All the mice were included in the current study. Animal experiments were approved and performed in accordance with the guidelines by Institutional Animal Care and Use Committee (IACUC) Sun Yat-sen University, Approval No. IACUC-DD-17-0604. Animal experiments were performed in accordance with the guidelines in Laboratory Animal Center of Sun Yat-sen University, Number of Animal Use Permit: SYSK (YU): 2011-0112. The mice were kept under controlled light conditions (12 h light-dark cycle) with free access to food and water. A standard diet for mice was provided by Laboratory Animal Center of Sun Yat-sen University. All efforts were made to minimize animal suffering.

After one week’ acclimatization, the 40 db/db mice were randomly distributed into five groups: (1) DM model group (n = 8); (2) DMMET130 group (n = 8, with metformin hydrochloride in a dose of 130 mg/kg); (3) DMDJP25 group (n = 8, with DJP in a dose of 25 mg/kg); (4) DMDJP50 group (n = 8, with DJP in a dose of 50 mg/kg); (5) DMDJP100 group (n = 6–8, with DJP in a dose of 100 mg/kg). The C57BL/6 mice were control group (n = 8), respectively.

All the drugs, DJP and metformin hydrochloride were given to the mice by gavage in 0.6% NaCMC solution. Non-treatment groups, DM model group and C57 control group, were given the same volume of 0.6% NaCMC solution to minimize the effects of gavage procedure. Two mice died for the reasons unrelated to the experiments in DMDJP100 group.

The drugs were administered once a day and for 8 consecutive weeks. Throughout the duration of trial, the blood glucose level and body weight of each mouse were monitored once a week. OGTT test was performed at the 7th week.

At the end of the trial, all the mice were sacrificed by cervical dislocation. The tissues were excised and frozen in liquid nitrogen, or 10% formalin solution immediately for further analysis. The contents of cecum and colon were sampled, weighed and immediately frozen in liquid nitrogen (46 in total), then, these gut samples were stored at −80 °C until DNA extraction.

### 4.4. Blood Glucose and ELISA Analysis 

Blood glucose level were determined on 3.5 μL of blood collected from the tip of the tail vein using a glucose meter, Accu-Chek Performa (Roche Diagnostics GmbH, Mannheim, Germany). OGTT test was performed as following: after fasting for 12 h, the mice were received 20% D-glucose solution by gavage in a dose of 2 g/kg, then, blood glucose level were measured at 0, 30, 60, 90, and 120 min following the glucose challenge.

The concentrations of TG, TC, HDL, LDL, INS and IL-6 in serum were determined using a commercial ELISA kit based on the manufacturer’s instructions. The levels of GSH, MDA, SOD and CAT in tissue samples from the mice kidney/liver were measured using a commercial ELISA kit based on the manufacturer’s instructions. A multi-mode microplate reader FlexStation 3 (Molecular devices, Sunnyvale, CA, USA) was used for ELISA testing.

### 4.5. Histologic Analysis

Freshly isolated tissues of liver as well as kidney were fixed overnight in 10% formalin solution, embedded in paraffin, cut into 3 μm thick sections, and stained with haematoxylin & eosin. The histological tissue damage of liver/kidney was observed using a LEICA DM5000B microscope (Leica, Heidelberg, Germany).

### 4.6. Immunohistochemical Analysis of TNF-α

Paraffin-embedded tissues sections (5 μm) were deparaffinized in xylene, rehydrated in a gradient of ethanol-water (85% ethanol, 5 min; 75% ethanol, 5min, then distilled water). Their antigen was repaired with EDTA antigen retrieval solution (pH 9.0); the endogenous peroxidase was quenched; the specific binding sections were blockaded by 3% bovine serum albumin. Then, the sections were incubated overnight at 4 °C with TNF-α antibody, with HRP conjugated anti-mouse IgG 50 mine, and were counterstained with hematoxylin. Images were taken under a FL Auto Imaging System+ EVOS^®^ Onstage Incubator (Life Technologies Corporation, CA, USA). The gray values were calculated using Image-Pro Plus 6.0 (Media Cybernetics, Inc., MA, USA).

### 4.7. 16S rRNA Gene Extraction, Amplification and Sequencing

The gut microbiome samples were sent to BGI Co., Ltd, China (Shenzhen, China) for DNA extraction and sequencing of 16S rRNA gene. Total genomic DNA of the gut microbiome was extracted using E.Z.N.A.^®^ Bacterial DNA Kit (Omega Bio-tek, Norcross, GA, USA) according to manufacturer’s instruction. The V4 of 16S rRNA gene from the single gut microbiome sample was amplified. After genome DNA was normalized to 30 ng per PCR reaction, V4 dual-index fusion PCR primer cocktail and PCR master mix were added, then to run PCR. The melting temperature was 56 °C and PCR cycle was 30. The PCR products were purified with AmpureXP beads to remove the unspecific products. The resulting library was used for sequencing on Illumina HiSeq 2500 platform following the standard pipelines of Illumina, and generating 2 × 250 bp paired-end reads [54].

### 4.8. Statistical and Bioinformatics Analysis 

To obtain clean reads, the raw data were filtered to eliminate the adapter pollution and low-quality reads by an in-house procedure as following [55]: The clean paired-end reads with overlap were merged to tags using FLASH (fast length adjustment of short reads, v1.2.11) [56]. Then, the tags were clustered to OTUs at 97% sequence similarity by scripts of software USEARCH (v9.1.13) [57]. Taxonomic ranks were assigned to OTUs representative sequence using Greengene (v201305, http://greengenes.secondgenome.com/downloads) [58]. At last, alpha diversity, beta diversity and the different species screening were analyzed based on OTUs and taxonomic ranks using mothur (v1.31.2), software R (v3.1.1), QIIME (v1.80), or metastats (http://metastats.cbcb.umd.edu/) [59,60].

### 4.9. Predicted Metabolic Profile

To predict the functional profile of the gut microbiome, the OTUs from the closed reference script were assigned to orthologous groups from the Kyoto Encyclopedia for Genes and Genomes using software PICRUSt [61,62]. 

## 5. Conclusions

Our study revealed that the rich-polyphenols extract of *D. loddigesii* (DJP) has the ability to improve the db/db mice symptoms of diabetes and complications, which might be the synergistic effects of all phenols of DJP. The anti-diabetic mechanism of DJP might be attributed to its anti- inflammation as well as anti-oxidatin and improving balance of intestinal flora. 

## Figures and Tables

**Figure 1 molecules-23-03245-f001:**
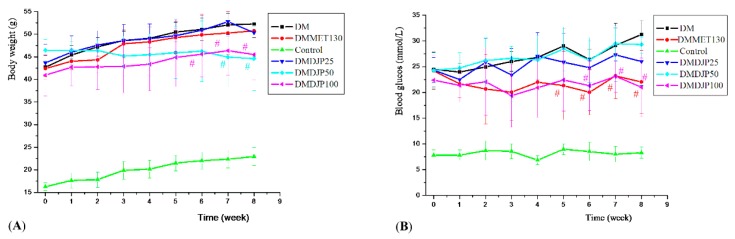
Effects of *D. loddigesii* (DJP) on body weight and blood glucose level across time in the mice. (**A**) Body weight changes over time; (**B**) Blood glucose level changes over time. Data are mean ± SD; # = *p* < 0.05 vs. DM group.

**Figure 2 molecules-23-03245-f002:**
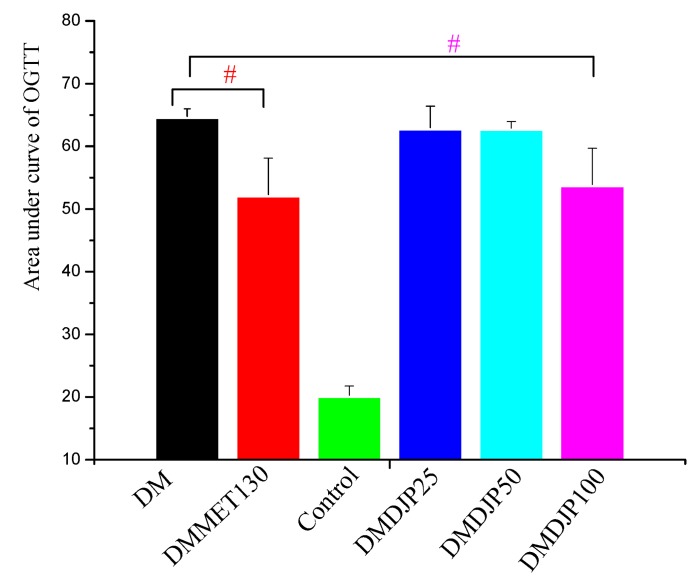
Effect of DJP on Oral Glucose Tolerance (OGTT). Data are mean ± SD; # = *p* < 0.05 vs. DM group.

**Figure 3 molecules-23-03245-f003:**
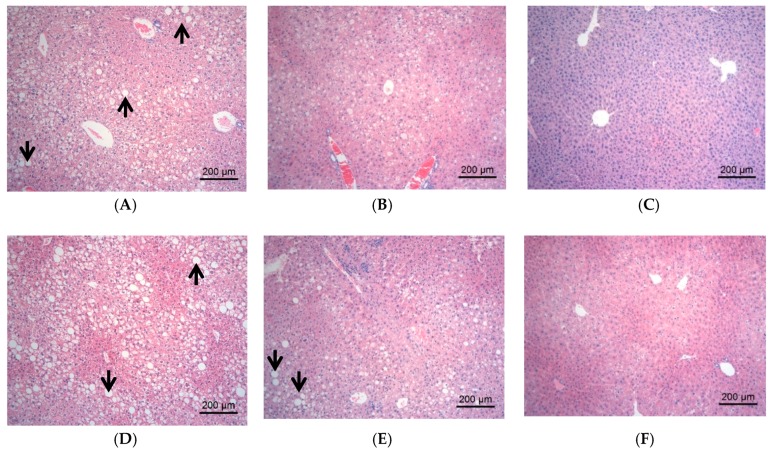
Pathological changes of liver tissues observed under light microscope. The representative areas of fatty degeneration of liver cells were indicated via black arrows, in which some fat droplets were observed under light microscope. (**A**) DM group, fatty degeneration of liver cells by approximately 80–90%; (**B**) DMMET130 group, fatty degeneration and balloon denaturalization of liver cells by approximately 30–40% and 10–20%; (**C**) Control group, normal morphology of liver cell; (**D**) DMDJP25 group, fatty degeneration of liver cells by approximately 70–80%; (**E**) DMDJP50 group, by approximately 40–50%; and (**F**) DMDJP100 group, by approximately 10–20%.

**Figure 4 molecules-23-03245-f004:**
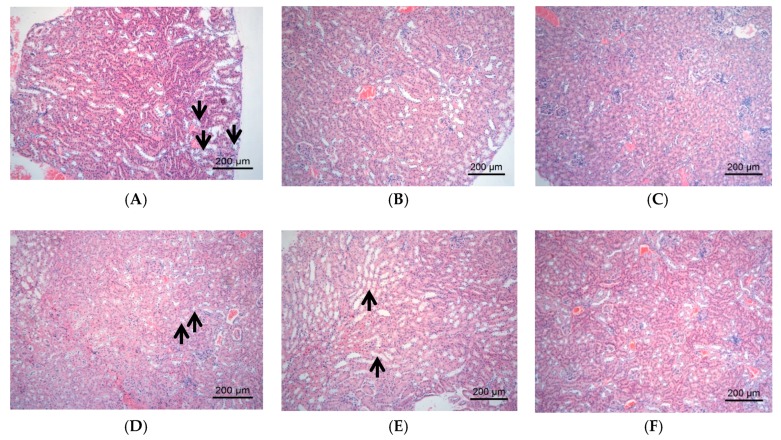
The pathological changes of kidney tissues observed under light microscope. The representative areas of vacuolar degeneration of renal tubular epithelial cells were indicated via black arrow, which appeared foam-like under light microscope. (**A**) DM group, vacuolar degeneration of renal tubular epithelial cells by approximately 21–40%; (**B**) DMMET130 group, normal renal cortex and medulla; (**C**) Control group, normal renal cortex and medulla; (**D**) DMDJP25 group, vacuolar degeneration by approximately 21–40%; (**E**) DMDJP50 group, vacuolar degeneration less than 20%; (**F**) DMDJP100 group, normal renal cortex and medulla.

**Figure 5 molecules-23-03245-f005:**
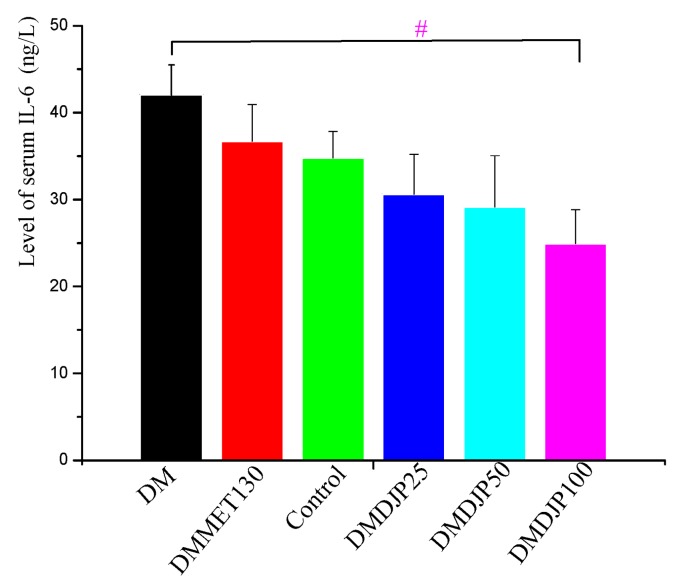
Level of serum IL-6 in the mice (ng/L). Data are mean ± SD; # = *p* < 0.05 vs. DM group.

**Figure 6 molecules-23-03245-f006:**
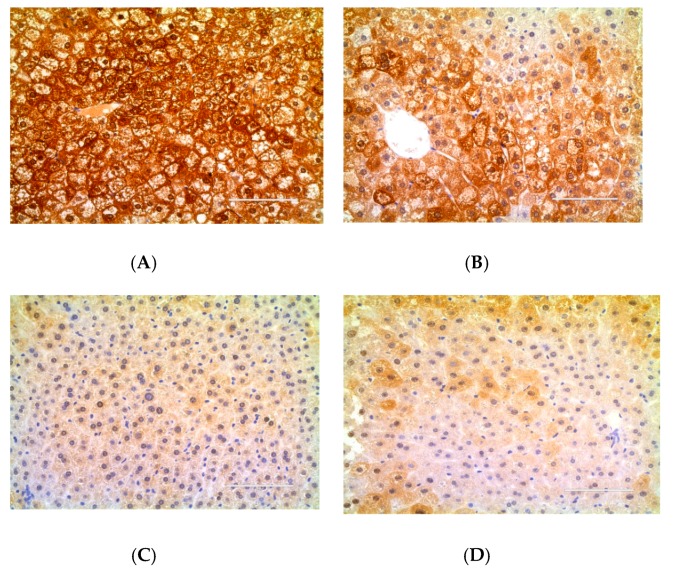
Expression of tumor necrosis factor-α (TNF-α) in the liver tissue detected by immunohistochemistry. The positive expressive areas of TNF-α are tan. (**A**) DM group; (**B**) DMMET130 group; (**C**) Control group; (**D**) DMDJP100 group.

**Figure 7 molecules-23-03245-f007:**
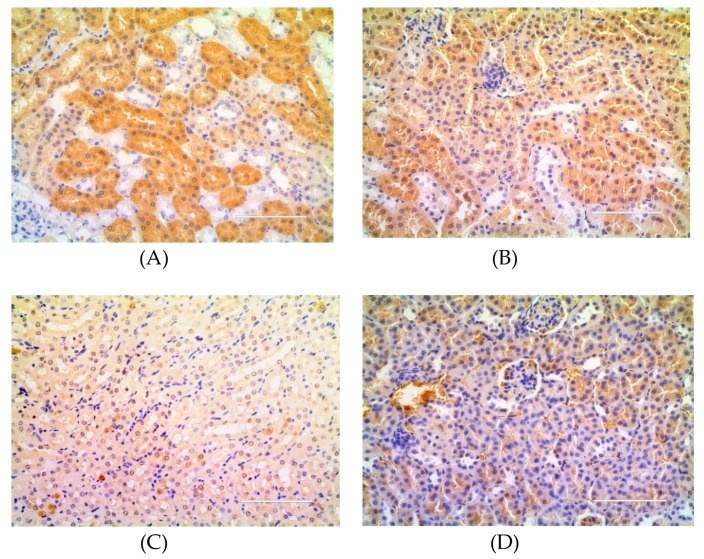
Expression of tumor necrosis factor-α (TNF-α) in kidney tissue detected by immunohistochemistry. The positive expressive areas of TNF-α are tan. (**A**) DM group; (**B**) DMMET130 group; (**C**) Control group; (**D**) DMDJP100 group.

**Figure 8 molecules-23-03245-f008:**
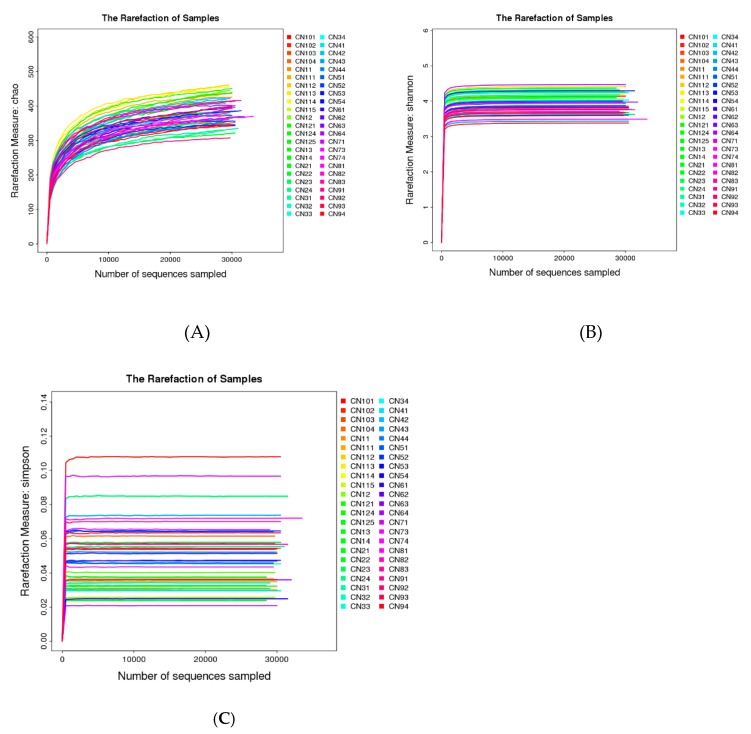
Rarefaction analysis curve based on alpha diversity. (**A**) Chao index rarefaction curve; (**B**) Shannon index rarefaction curve; (**C**) Simpson index rarefaction curve.

**Figure 9 molecules-23-03245-f009:**
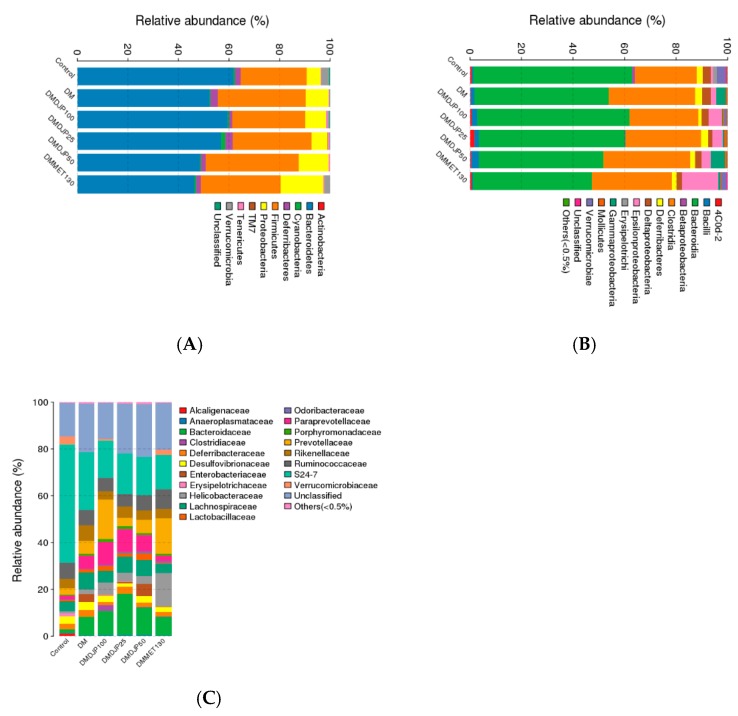
Taxonomic composition of gut microbiome in the mice. (**A**) Phylum-level; (**B**) Class-level; (**C**) Species-level.

**Figure 10 molecules-23-03245-f010:**
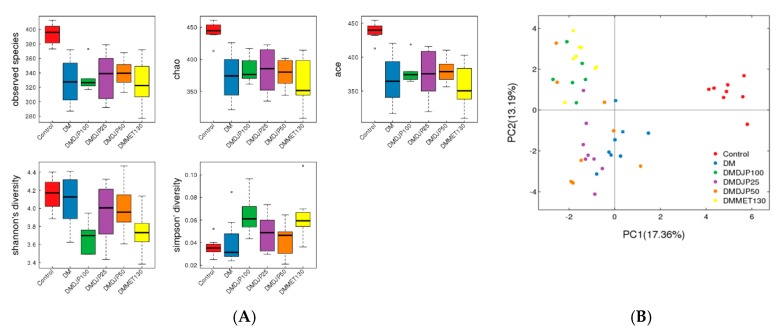
Alpha diversity indices box-plots and principal component analysis (PCA) plots based on operational taxonomic units (OTUs) abundance. (**A**) box-plots; (**B**) PCA plots

**Figure 11 molecules-23-03245-f011:**
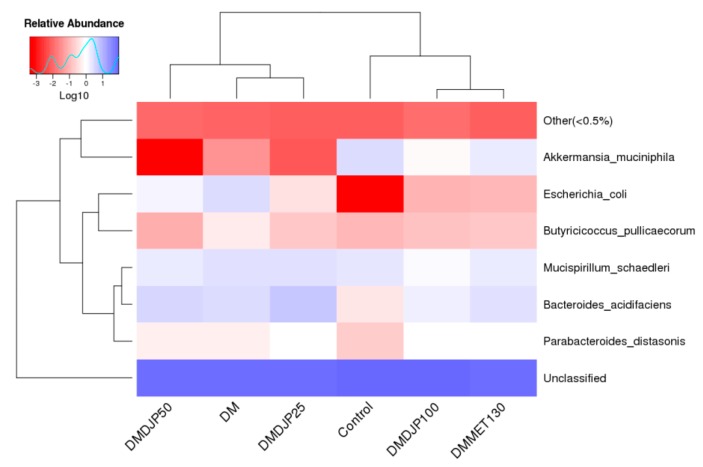
The heat map of species-level of the mice.

**Figure 12 molecules-23-03245-f012:**
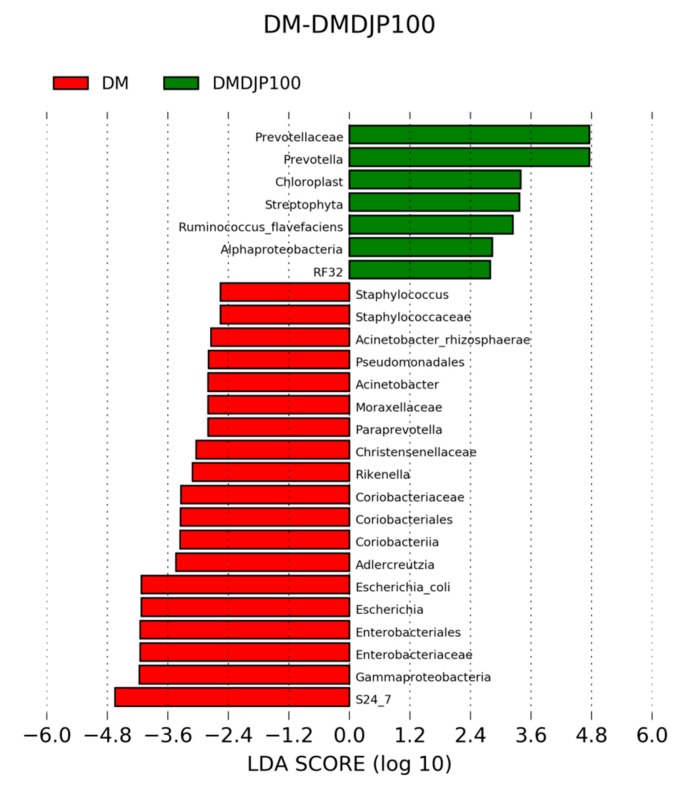
Identification of taxonomic features relevant in DM and DMDJP100 using LEfSe (LDA Effect Size).

**Table 1 molecules-23-03245-t001:** Effects of *D. loddigesii* (DJP) on triglyceride (TG), total cholesterol (TC), high density lipoprotein cholesterol (HDL-C), low density lipoprotein cholesterol (LDL-C), and insulin (INS) levels in the mice serum.

Group	TG	TC	HDL-C	LDL-C	INS
	mmol/L				mIU/L
DM	1.45 ± 0.21	5.06 ± 0.49	0.99 ± 0.18	1.19 ± 0.21	7.06 ± 1.63
DMMET130	1.32 ± 0.29	4.69 ± 0.59	1.01 ± 0.12	1.20 ± 0.12	11.18 ± 1.82 ^#^^, 1^
Control	1.01 ± 0.27	2.51 ± 0.58	1.10 ± 0.21	0.65 ± 0.10	10.64 ± 3.74
DMDJP25	1.50 ± 0.31	4.98 ± 0.69	0.93 ± 0.12	1.25 ± 0.11	8.22 ± 3.80
DMDJP50	1.48 ± 0.19	4.93 ± 0.59	0.96 ± 0.19	0.69 ± 0.20 ^#^	10.93 ± 3.48 ^#^
DMDJP100	1.49 ± 0.33	4.48 ± 0.25	1.03 ± 0.12	0.65 ± 0.11 ^#^	11.43 ± 3.37 ^#^

Data are means ± SD; ^1^: ^#^ = *p* < 0.05 vs. DM group.

**Table 2 molecules-23-03245-t002:** Grading of liver fatty degeneration/vacuolar degeneration of renal tubule epithelium cells in the mice.

Group	N	Grading of Liver Fatty Degeneration						Grading of Vacuolar Degeneration of Renal Tubule Epithelium Cells		
		0	+	++	+++	++++	+++++	0	++	+++
DM	8	0	0	0	0	5	3	0	6	2
DMMET130 *^, 1^	8	3	0	0	1	2	2	8	0	0
Control	8	8	0	0	0	0	0	8	0	0
DMDJP25	8	1	0	0	2	4	1	1	4	3
DMDJP50 ^#^^, 2^	8	3	2	0	1	2	0	2	4	2
DMDJP100 ^##,^ ^3,^ *	6	1	1	2	2	0	0	6	0	0

Date in Table 2 are the number of animals; Grading: 0 = No change of histopathology; + = less than 10% change of histopathology; ++ = less than 25% change of histopathology; +++ = 25–50% change of histopathology; ++++ = 50–75% change of histopathology; +++++ = > 75% change of histopathology; ^1^ * = *p* < 0.01 vs. DM group (renal tissue); ^2^
^#^ = *p* < 0.05 vs. DM group (liver tissue); ^3^
^##^ = *p* < 0.01 vs. DM group (liver tissue).

**Table 3 molecules-23-03245-t003:** Immunohistochemical expression of TNF-α of liver/ kidney tissue in the mice.

Group	Expression of TNF-α in Liver Tissue	Expression of TNF-α in Kidney Tissue
DM	0.21 ± 0.03	0.077 ± 0.005
DMMET130	0.18 ± 0.05	0.110 ± 0.002
Control	0.069 ± 0.006	0.028 ± 0.003
DMDJP100	0.088 ± 0.005 ^##^^, 1^	0.034 ± 0.003 ^#^^, 2^

The data were mean of gray scale (IOD/area); ^1^
^##^ = *p* < 0.01 vs. DM group, ^2^
^#^ = *p* < 0.05 vs. DM group.

**Table 4 molecules-23-03245-t004:** Oxidative stress index of liver/kidney tissue in the mice.

Tissue	Oxidative Stress Index		Group			
			DM	DMMET130	Control	DMDJP100
Liver	GSH	μmol/gprot	37.06 ± 2.87	40.58 ± 2.61 ^#^^, 1^	44.51 ± 3.70	41.86 ± 3.43 ^#^
	MDA		0.66 ± 0.03	0.34 ± 0.05 ^##, 2^	0.59 ± 0.08	0.44 ± 0.05 ^##^
	CAT	U/mgprot	45.92 ± 4.41	57.48 ± 4.27 ^##^	47.21 ± 1.59	70.35 ± 5.07 ^##^
	SOD		161.5 ± 5.9	161.7 ± 7.5	205.8 ± 8.7	177.4 ± 6.4 ^#^
Kidney	GSH	μmol/gprot	27.88 ± 2.26	37.82 ± 2.72 ^#^	36.27 ± 3.88	33.38 ± 3.09 ^#^
	MDA		1.79 ± 0.08	1.38 ± 0.13 ^##^	1.31 ± 0.15	1.51 ± 0.11 ^#^
	CAT	U/mgprot	12.43 ± 5.23	30.48 ± 5.49 ^##^	30.11 ± 3.34	32.21 ± 4.53 ^##^
	SOD		83.65 ± 6.8	115.2 ± 6.3 ^##^	131.1 ± 5.8	111.0 ± 5.6 ^#^

Data are means ± SD; ^1^
^#^ = *p* < 0.05 vs. DM group; ^2 ##^ = *p* < 0.01 vs. DM group.

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
