# Peer review of "Effects of Rich-Polyphenols Extract of Dendrobium loddigesii on Anti-Diabetic, Anti-Inflammatory, Anti-Oxidant, and Gut Microbiota Modulation in db/db Mice"

_molecules, 2018, doi:10.3390/molecules23123245_

Round 1
Reviewer 1 Report
- Figure S1 is missing (polyphenol structure) and should be added
- The major polyphenol component represent roughly 10% of the extract, do you have an idea of the 90% remaining (for a better extract characterization)
- Is the efficient dosage (100mg/kg) used in mice is representative of the traditional use in man ? It seems to be a very high dosage, far away of what could be used in human nutrition/drug
- Table 1 : the total cholesterol (TC) does not correspond to the sum of HDL + LDL-C : do you have an explanation ?
- Oxidative stress (table 4) : Liver GSH level is surprinsigly higher in the DM group / control, we should expect a lower value (as in the kidney) as the DM group is under high oxidative stress : have you double checked the result, or have an explanation ?
- Anti inflammatory effect (table 5) : the extract shows an anti inflammatory effect according to the IL-6 reduction, but surprinsigly the control group is more or less at the same level than the DM group : do you have an explanation ?
Author Response
Title: Effects of Dendrobium loddigesii polyphenols on anti-diabetic, anti-oxidant, anti-inflammatory and gut microbiota modulation in db/db mice
Journal: molecules
Dear expert:
Thank you for the useful comments and suggestions on our manuscript. We have modified the manuscript accordingly, and the detailed corrections are listed below point by point.
Comment (1): Figure S1 is missing (polyphenol structure) and should be added
Response: Thanks for your directions. The missing polyphenol structures have been added in Figure S1: “The others polyphenol structures are reported in the work, Li, X.W.; Chen, H.P.; He, W.B.; Yang, W.L.; Ni, F.Y.; Huang, Z.W.; Hu, H.Y.; Wang, J. Polyphenols from Dendrobium loddigesii and their biological activities. Acta Sci. Nat. Univ. Sunyatseni accepted)”.
21 polyphenols were identified and their structures were listed following:
Comment (2): The major polyphenol component represent roughly 10% of the extract, do you have an idea of the 90% remaining (for a better extract characterization)
Response: Thanks for your directions. “D. loddigesii polyphenols” has been replaced by “rich-polyphenols extract of D. loddigesii” according to your advice.
There was no good idea for the 90% remaining now. But I thought the rich-polyphenols extract may contain mainly polyphenols and polysaccharides. Its enrichment method might get rid of some polysaccharides (H2O-eluent), alkaloids shihunine (H2O-eluent), fatty acids (80-100% methanol eluent) and esters (80-100% methanol eluent). The following were 1H NMR of enrichment process:
1H NMR of H2O-eluent 1H NMR of 80 % methanol eluent
1H NMR of 90 % methanol eluent 1H NMR of 100 % methanol eluent
Comment (3): Is the efficient dosage (100mg/kg) used in mice is representative of the traditional use in man? It seems to be a very high dosage, far away of what could be used in human nutrition/drug
Response: Thanks for your directions. The dosage of 100mg/kg used was designed based on the enrichment of rich-polyphenols extract and the literature value. The literature values of crude extract were usually 300−500 mg/kg; the enrichment of rich-polyphenols extract was 3: the total weight of H2O-eluent was one third of the crude extract and that of 80−100 % methanol eluent was one third too. So the dosage of 100mg/kg was used in the experiments.
The sentence “Our results suggested that the rich-polyphenols extract from D. loddigesii may be used as adjuvant therapy to diabetes and its complications” has been deleted according to your advice. Indeed, much work will be needed to do.
Comment (4): Table 1: the total cholesterol (TC) does not correspond to the sum of HDL + LDL-C : do you have an explanation ?
Response: Thanks for your directions. The total cholesterol (TC) might include other free cholesterols or cholesterol esters. I am so sorry that I'm not good at it.
Comment (5): Oxidative stress (table 4): Liver GSH level is surprinsigly higher in the DM group /control, we should expect a lower value (as in the kidney) as the DM group is under high oxidative stress: have you double checked the result, or have an explanation?
Response: Thanks for your directions. Liver GSH level was retested for correcting any errors; others data have been recalculated; the mistakes found have been corrected. I' m terribly sorry. Thanks you once again
Table 4. Oxidative stress index of liver/kidney tissue in the mice
Tissue | Oxidative stress index | Group | |||||||||
DM | DMMET130 | Control | DMDJP100* | ||||||||
Liver | GSH | umol/gprot | 37.06 ± 2.87 | 40.58 ± 2.61# | 44.51 ± 3.70 | 41.86 ± 3.43 # | |||||
Comment (6): Anti inflammatory effect (table 5) : the extract shows an anti inflammatory effect according to the IL-6 reduction, but surprinsigly the control group is more or less at the same level than the DM group : do you have an explanation ?
Response: Thanks for your directions. The IL-6 level of control group and DM group were retested for correcting any errors; others data have been recalculated; the mistakes found have been corrected. I' m terribly sorry. Thanks you once again.
Figure 5. Level of serum IL-6 in the mice (ng/L). Data are mean ± SD; # = P < 0.05 vs. DM group.
Thank you very much once again.
Sincerely yours,
Ph.D. Jun Wang
School of Pharmaceutical Sciences
Sun Yat-sen University
Guangzhou, China, 510006
Phone: +86-20-39943090
Fax: +86-20-39943090
E-mail: wjun@mail.sysu.edu.cn

Reviewer 2 Report
The manuscript is the result of a well planned and carried out experiments.
The authors conducted a series of tests and analyzes based on valuable research methodologies. The work can interest many readers and provide them with interesting information.
In figure 3 and figure 4, a representative area of changes should be marked.
Figure 11 is illegible and needs improvement.
Author Response
Title: Effects of Dendrobium loddigesii polyphenols on anti- diabetic, anti-oxidant, anti-inflammatory and gut microbiota modulation in db/db mice
Journal: molecules
Dear expert:
Thank you for the useful comments and suggestions on our manuscript. We have modified the manuscript accordingly, and the detailed corrections are listed below point by point.
Comment (1): In figure 3 and figure 4, a representative area of changes should be marked.
Response: Thanks for your directions. In figure 3 and figure 4, some short explanations have been added:
Figure 3. Pathological changes of liver tissues observed under light microscope. The representative areas of fatty degeneration of liver cells were indicated via black arrows, in which some fat droplets were observed under light microscope;
Figure 4. The pathological changes of kidney tissues observed under light microscope. The representative areas of vacuolar degeneration of renal tubular epithelial cells were indicated via black arrow, which appeared foam-like under light microscope. Thank you very much.
Comment (2): Figure 11 is illegible and needs improvement.
Response: Thanks for your directions. In order to get a clear features image, Figure 11 has been changed as Figure S2 (original size) and into Supplementary Materials. I am so sorry that my drawing level was too low to modify Figure 11, which was generated by statistical software based on the data.
Thank you very much once again.
Sincerely yours,
Ph.D., Jun Wang
School of Pharmaceutical Sciences
Sun Yat-sen University
Guangzhou, China, 510006
Phone: +86-20-39943090
Fax: +86-20-39943090
E-mail: wjun@mail.sysu.edu.cn

Reviewer 3 Report
The author submitted the MS “Effects of Dendrobium loddigesii polyphenols on antidiabetic, anti-oxidant, anti-inflammatory and gut microbiota modulation in db/db mice. By Hue-wen Li, Hui-ping Chen, Ying-yan He, Wei-li Chen, Jian-wen Chen, Lu Gao, Hai-yan Hu, Jun Wang” for possible publication at the Journal. After the review, I suggest to publish at the Journal after minor revision. English grammar and some improvement of style, and re-writing of results where necessary. In Introduction described that “Type 2 diabetes mellitus (T2DM) is a growing threat to public health and the correlated with metabolism and diseases. It produces hyperglycemia, insulin resistance and decrease of insulin secretion. It causes death and disability, due to various severe and whole body complications, such as, heart, kidneys, liver, brain and other organs.” Therefore, their aim is “To study anti-diabetic effects of Dendrobium loddigesii polyphenols, which was prepared as ethanol extract and its anti-diabetic effects, antioxidant/anti-inflammatory activities, and modulation of gut microbiota were studied using db/db mice.” Since “The Chinese traditional medicine clinic uses Dendrobium plants as ethanol extract, to treat T2DM/complications, which ameliorates diabetic retinopathy. The D. loddigesii crude drugs, called “Shi Hu”, contains total polyphenols (DJP). “The authors used total polyphenols “To understand the anti-diabetic effect of DJP, on the oxidative stress parameters, inflammatory factors and modulation of gut microbiota, studied in the db/db mice.” The results showed that DJP polyphenols improved the symptoms of diabetes/complications in db/db mice, decreased the inflammation factors/oxidative stress and improved intestinal flora balance.
I have some suggestion to the authors:
- I suggest adding an Abbreviation list in the MS. In addition, checking define DM
- In Line 77, delete “This paper reported on the result of study”
In Line 79-80, change the phrase “The stems of Dendrobium plants were used to treat T2DM/complications with a long history in Chinese, generally using in Chinese medicine prescript.” By “In Chinese medicine prescript uses Dendrobium plants to treat T2DM/complications.”
In Line 89, explain why “metformin (MET) treatment group (in a dose of 130 mg/kg).” uses as…
In Line 107, Change “Figure 1A showed that the body weight of db/db mice was very much heavier than C57 control mice.” By “Figure 1A showed that the body weight of db/db mice was heavier than C57 control mice” indicate the percentage of body weight increases.
In Line 109, Change “the body weight of DMDJP50 and DMDJP100 groups were significantly decreased after 7 or 6 weeks of treatment” By “After 6 or 7 weeks of DMDJP50 and DMDJP100 treatment, the body weight of db/db mice decreased significantly, by X %, Y %, respectively.
In Line 111, change the phrase “Figure 1B and Figure 2 displayed that the blood glucose level and OGTT test in the db/db mice were obviously higher than that C57 control mice.” By “Figure 1B showed that as DJP concentrations increases up to DMDJP100 the blood glucose level enhanced to X mmol/L compared to mice C57 control, Y mmol/L. Figure 2 displayed that the OGTT test in the db/db mice indicated that all DJP concentrations tested showed higher OGTT area (indicating higher level of glucose?) than mice C57 control by enhancing glucose by X %.”
In Line 112, change the phrase “There were no any improvements of the blood glucose level and OGTT test in the mice of DMDJP25 and DMDJP50 groups.” By “However, the level of blood glucose and the OGTT test in the db/db mice were not affected with concentrations of DMDJP25 and DMDJP50 tested.”
In Line 113, change the phrase “The blood glucose level/OGTT test in DMMET130 group were significantly improved after 5 weeks of treatment (P < 0.05 vs. DM group)” By “DMMET130 group enhanced the blood glucose level by X % in OGTT test after 5 weeks of treatment (P < 0.05) compared to DM group”
In Line 115, use this phrase “DMDJP100 group enhanced the blood glucose level in OGTT test (P < 0.05 vs. DM group) after 6 weeks of treatment, which was similarly to that effect induced by DMMET130 group.”
In Line 124 to 129, change the phrase “Table 1 indicated that the concentration of TG and TC in the db/db mice serum is obviously higher than that of C57 control mice; while the TG, TC and HDL-C levels were not improved after the treatment with different doses of DJP or MET (in 130 mg/kg dose). The LDL-C levels in C57 control, DMDJP50 and DMDJP100 groups were obviously lower than that of DM group (P < 0.05), which were not improved in DMDJP25 and DMMET130 groups. The concentration of INS in DM group was lower than that of control group or the treated groups; the serum INS level in DMDJP50, DMDJP100 and DMMET130 groups were obviously higher than that of DM group (P < 0.05).“ By “Table 1 indicated that DMDJP25, DMDJP50 and DMDJP100 treatments of db/db mice, enhanced the concentration of TG and TC in serum by approximately 1.50 mmol/L compared to control C57 mice (1.01 mmol/L). In addition, in the same experiments TG, TC and HDL-C levels were not affected with different doses of DJP or MET (in 130 mg/kg dose). Further experiments in the same mice with DMDJP50 and DMDJP100 the level of LDL-C in C57 control, decreased than DM group (P < 0.05) by X mmol/L, which were not enhanced with DMDJP25 and DMMET130 treatment. Furthermore, the concentration of INS in DM group was lower by Y mmol/L than control group or DJP treatment. Finally, the serum INS level in the db/db mice treated with DMDJP50, DMDJP100 and DMMET130 increased by Z mmol/L than DM group N mmol/L (P < 0.05)”
Author Response
Title: Effects of Dendrobium loddigesii polyphenols on anti- diabetic, anti-oxidant, anti-inflammatory and gut microbiota modulation in db/db mice
Journal: molecules
Dear expert:
Thank you for the useful comments and suggestions on our manuscript. We admire your excellent academic level and responsibility. You are so kind and go over the manuscript word for word to correct and improve it; I am very thank you. We have modified the manuscript accordingly, and the detailed corrections are listed below point by point.
Comment (1): I suggest adding an Abbreviation list in the MS. In addition, checking define DM
Response: Thanks for your directions. The Abbreviation list has been added as per your advice. The define of DM has been added in Abbreviation list.
Comment (2): In Line 77, delete “This paper reported on the result of study”
Response: Thanks for your directions. The sentence has been deleted.
Comment (3): Line 79-80, change the phrase “The stems of Dendrobium plants were used to treat T2DM/complications with a long history in Chinese, generally using in Chinese medicine prescript.” By “In Chinese medicine prescript uses Dendrobium plants to treat T2DM/complications.”
Response: Thanks for your directions. The phrase has been replaced.
Comment (4): In Line 89, explain why “metformin (MET) treatment group (in a dose of 130 mg/kg).” uses as….
Response: Thanks for your directions. The objective of establishing metformin group has been explained: metformin (MET) treatment group (DMMET130 group, in a dose of 130 mg/kg) was used as a positive control group.
Comment (5): In Line 107, Change “Figure 1A showed that the body weight of db/db mice was very much heavier than C57 control mice.” By “Figure 1A showed that the body weight of db/db mice was heavier than C57 control mice” indicate the percentage of body weight increases.
Response: Thanks for your directions. The phrase has been replaced; the percentage of body weight increases has been indicated: 128−98.7%.
Comment (6): In Line 109, Change “the body weight of DMDJP50 and DMDJP100 groups were significantly decreased after 7 or 6 weeks of treatment” By “After 6 or 7 weeks of DMDJP50 and DMDJP100 treatment, the body weight of db/db mice decreased significantly, by X %, Y %, respectively.
Response: Thanks for your directions. The phrase has been replaced; the percentage of body weight decrease has been indicated: about 15% and 13%, respectively.
Comment (7): In Line 111, change the phrase “Figure 1B and Figure 2 displayed…… were obviously higher than that C57 control mice.” By “Figure 1B showed that as DJP concentrations increases up to DMDJP100 the blood glucose level enhanced to X mmol/L compared to mice C57 control, Y mmol/L, Figure 2 displayed that the OGTT test in the db/db mice indicated that all DJP concentrations tested showed higher OGTT area (indicating higher level of glucose?) than mice C57 control by enhancing glucose by X %.”
Response: Thanks for your directions. The phrase has been replaced; the blood glucose values have been added.
Comment (8): In Line 112, change the phrase “There were no any improvements of the blood glucose level and OGTT test in the mice of DMDJP25 and DMDJP50 groups.” By “However, the level of blood glucose and the OGTT test in the db/db mice were not affected with concentrations of DMDJP25 and DMDJP50 tested.”
Response: Thanks for your directions. The phrase has been replaced.
Comment (9): In Line 113, change the phrase “The blood glucose level/OGTT test in DMMET130 group were significantly improved after 6 weeks of treatment (P < 0.05 vs. DM group)” By “DMMET130 group enhanced the blood glucose level by X % in OGTT test after 6 weeks of treatment (P < 0.05) compared to DM group”
Response: Thanks for your directions. The phrase has been replaced; the values have been added
Comment (10): In Line 115, use this phrase “DMDJP100 group enhanced the blood glucose level in OGTT test (P < 0.05 vs. DM group) after 6 weeks of treatment, which was similarly to that effect induced by DMMET130 group.”
Response: Thanks for your directions. The phrase has been replaced.
Comment (11): Line 124 to 129, change the phrase “Table 1 indicated that the concentration of TG and TC in the db/db mice serum is obviously higher than that of C57 control mice; while the TG, TC and HDL-C levels were not improved after the treatment with different doses of DJP or MET (in 130 mg/kg dose). The LDL-C levels in C57 control, DMDJP50 and DMDJP100 groups were obviously lower than that of DM group (P < 0.05), which were not improved in DMDJP25 and DMMET130 groups. The concentration of INS in DM group was lower than that of control group or the treated groups; the serum INS level in DMDJP50, DMDJP100 and DMMET130 groups were obviously higher than that of DM group (P < 0.05).“ By “Table 1 indicated that DMDJP25, DMDJP50 and DMDJP100 treatments of db/db mice, enhanced the concentration of TG and TC in serum by approximately 1.50 mmol/L compared to control C57 mice (1.01 mmol/L). In addition, in the same experiments TG, TC and HDL-C levels were not affected with different doses of DJP or MET (in 130 mg/kg dose). Further experiments in the same mice with DMDJP50 and DMDJP100 the level of LDL-C in C57 control, decreased than DM group (P < 0.05) by X mmol/L, which were not enhanced with DMDJP25 and DMMET130 treatment. Furthermore, the concentration of INS in DM group was lower by Y mmol/L than control group or DJP treatment. Finally, the serum INS level in the db/db mice treated with DMDJP50, DMDJP100 and DMMET130 increased by Z mmol/L than DM group N mmol/L (P < 0.05)”
Response: Thanks for your directions. The phrase has been replaced; the values have been added.
Thank you very much once again.
Sincerely yours,
Ph.D., Jun Wang
School of Pharmaceutical Sciences
Sun Yat-sen University
Guangzhou, China, 510006
Phone: +86-20-39943090
Fax: +86-20-39943090
E-mail: wjun@mail.sysu.edu.cn

Reviewer 4 Report
The manuscript entitled “Effects of Dendrobium loddigesiipolyphenols on anti-diabetic, anti-oxidant, anti-inflammatory and gut microbiota modulation in db/db mice” examined that the fraction obtained from Dendrobium loddigesiishows beneficial effects against diabetes related phenotypes in db/db mice. Moreover, gut microbiome in the mice was changed by treatment of the fraction containing some phenols.
The study provides important information to know the effects of the fraction obtained from Dendrobium loddigesii. The fraction contains polyphenols such as moscatilin that are reported to have an anti-inflammatory activity.
Comment:
1). What are the purities of diversity phenols in the DJP fraction used in this study? Page 3, line 93-96, The DJP fraction contains diversity phenols such as moscatilin, gigantol, 2,4,7-trihydroxyl-9,10-dihydrophenanthrene, and tristin. Total concentration of these phenols is less than 10%. Therefore, it is difficult to mention the effects of these phenols by using the DJP fraction. If so, the title and message of this study are not appropriate.
2). How were the expression levels of TNF-α determined in Figure 6, Figure 7, and Table 3? From the explanation of the Results section, it cannot be understood that the results in Table 3 are obtained from Figure 6 and Figure 7. For example, in Figure 7, the number of purple dots of picture D (DMDJP100) is more than that of picture C (Control). In addition, in Figure 7, the yellow area of picture D (DMDJP100) is more than that of picture C (Control). However, in Table 3, the expression of TNF-α of DMDJP100 is lower than that of Control.
3). The authors should write the Figure legends. For example, in Figure 8, it is difficult to understand from the Figure legend and the Results section. What is the meaning of chao, shannon, and simpson in Figure 8? The appropriate information should be added in the Figure legends in the manuscript.
4). There are many long sentences throughout the article, and the sentences are hard to understand. The long sentences should be altered to shorten sentences. For example, sentences (page 1, lane 17-22, and line 22-27) in Abstract section are long.
3). The authors should check again in whole manuscript to improve the minor miss-descriptions. For example, “in vitro” in page 2, lane 74, and Supplementary Table 1, and “D. loddigesii” in Supplementary Figure 1 should be written in Italic.
What is the “diabetes/complications”? The meaning is “diabetes and complications caused by diabetes”?
Author Response
Title: Effects of Dendrobium loddigesii polyphenols on anti-diabetic, anti-oxidant, anti-inflammatory and gut microbiota modulation in db/db mice
Journal: molecules
Dear expert:
Thank you for the useful comments and suggestions on our manuscript. We have modified the manuscript accordingly, and the detailed corrections are listed below point by point.
Comment (1): What are the purities of diversity phenols in the DJP fraction used in this study?
Response: Thanks for your directions. The purities of all diversity phenols in the DJP fraction were not analysed; only four component contents were evaluated. Other trace phenols in the DJP fraction were obtained with chromatographic techniques, which contents were too low to be analysed at our experiment conditions. 21 polyphenols were identified from DJP fraction and their structures were listed following:
Comment (2): Page 3, line 93-96, The DJP fraction contains diversity phenols such as moscatilin, gigantol, 2,4,7-trihydroxyl-9,10-dihydrophenanthrene, and tristin. Total concentration of these phenols is less than 10%. Therefore, it is difficult to mention the effects of these phenols by using the DJP fraction. If so, the title and message of this study are not appropriate.
Response: Thanks for your directions. The title has been revised as “Effects of rich-polyphenols extract of Dendrobium loddigesii on anti-diabetic, anti-inflammatory anti-oxidant and gut microbiota modulation in db/db mice” according to your advice; other inaccuracy messages have been revised in manuscript, such as, the sentence “which are different from flavonoids and resveratrols (Line 63 )” has been deleted; the conclusions have been revised as “Our study revealed that the rich-polyphenols extract of D. loddigesii (DJP) has the ability to improve the db/db mice symptoms of diabetes and complications, which might be the synergistic effects of all phenols of DJP.” Thank you once again.
Comment (3): How were the expression levels of TNF-α determined in Figure 6, Figure 7, and Table 3?
Response: Thanks for your directions. Table 3 was obtained from the IOD/area of Figure 6 and Figure 7.
Comment (4): From the explanation of the Results section, it cannot be understood that the results in Table 3 are obtained from Figure 6 and Figure 7. For example, in Figure 7, the number of purple dots of picture D (DMDJP100) is more than that of picture C (Control). In addition, in Figure 7, the yellow area of picture D (DMDJP100) is more than that of picture C (Control). However, in Table 3, the expression of TNF-α of DMDJP100 is lower than that of Control.
Response: Thanks for your directions. I' m terribly sorry. Expression level of TNF-α in kidney tissue of control group in Table 3 was mistaken, which has been corrected. Thank you once again.
Table 3. Immunohistochemical expression of TNF-α of liver/ kidney tissue in the mice
Group | Expression of TNF-α in liver tissue | Expression of TNF-α in kidney tissue |
Control | 0.069 ± 0.006 | 0.028 ± 0.003 |
DMDJP100, | 0.088 ± 0.005## | 0.034 ± 0.003# |
Comment (5): The authors should write the Figure legends. For example, in Figure 8, it is difficult to understand from the Figure legend and the Results section. What is the meaning of chao, shannon, and simpson in Figure 8? The appropriate information should be added in the Figure legends in the manuscript.
Response: Thanks for your directions. The figure legends of Figure 3, Figure 4, Figure 6, Figure 7 and Figure 8 have been added.
Figure 3. Pathological changes ..…. light microscope. The representative areas of fatty degeneration of liver cells were indicated via black arrows, in which some fat droplets were observed under light microscope.
Figure 4. The pathological changes …… light microscope. The representative areas of vacuolar degeneration of renal tubular epithelial cells were indicated via black arrow, which appeared foam-like under light microscope.
Figure 6. Expression of TNF-α …… immunohistochemistry. The positive expressive areas of TNF-α are tan.
Figure 7. Expression of TNF-α …… immunohistochemistry. The positive expressive areas of TNF-α are tan.
Figure 8. Rarefaction analysis curve based on alpha diversity. A: Chao index rarefaction curve; B: Shannon index rarefaction curve; C: Simpson index rarefaction curve.
The information about Figure 8 has been revised in the manuscript: The sequencing quality was evaluated by methods of rarefaction analysis based alpha diversity indexes (Figure 8). From Figure 8, we can see three rarefaction curves (Figure 8A Chao1 index curve, 8B Shannon index curve and 8C Simpson index curve) tend to be flat or reach the plateau stage, which suggest the sequencing data were enough to cover all species in the microbiome community, and the most of the gut microbial diversity in each sample was captured with the current sequencing depth.
Comment (7): There are many long sentences throughout the article, and the sentences are hard to understand. The long sentences should be altered to shorten sentences. For example, sentences (page 1, lane 17-22, and line 22-27) in Abstract section are long.
Response: Thanks for your directions. The long sentences have been altered to shorten sentences according to different activities.
Comment (8): The authors should check again in whole manuscript to improve the minor miss-descriptions. For example, “in vitro” in page 2, lane 74, and Supplementary Table 1, and “D. loddigesii” in Supplementary Figure 1 should be written in Italic.
Response: Thanks for your directions. The “in vitro” and “D. loddigesii” in manuscript or supplementary have been corrected into italic.
Comment (9): What is the “diabetes/complications”? The meaning is “diabetes and complications caused by diabetes”?
Response: Thanks for your directions. “diabetes/complications” has been corrected into diabetes and complications according to your advice.
Thank you very much once again.
Sincerely yours,
Ph.D. Jun Wang
School of Pharmaceutical Sciences
Sun Yat-sen University
Guangzhou, China, 510006
Phone: +86-20-39943090
Fax: +86-20-39943090
E-mail: wjun@mail.sysu.edu.cn

Round 2
Reviewer 4 Report
The authors carefully responded to my comments and revised the manuscript. The responses to my comments are appropriate. The revised manuscript is highly improved.